# Vaccine Efficacy of a Newly Developed Feed-Based Whole-Cell Polyvalent Vaccine against Vibriosis, Streptococcosis and Motile Aeromonad Septicemia in Asian Seabass, *Lates calcarifer*

**DOI:** 10.3390/vaccines9040368

**Published:** 2021-04-10

**Authors:** Aslah Mohamad, Mohd Zamri-Saad, Mohammad Noor Azmai Amal, Nurhidayu Al-saari, Md. Shirajum Monir, Yong Kit Chin, Ina-Salwany Md Yasin

**Affiliations:** 1Aquatic Animal Health and Therapeutics Laboratory, Institute of Bioscience, Universiti Putra Malaysia, Serdang 43400, Selangor, Malaysia; gs51699@student.upm.edu.my (A.M.); mzamri@upm.edu.my (M.Z.-S.); mnamal@upm.edu.my (M.N.A.A.); gs57258@student.upm.edu.my (Y.K.C.); 2Department of Veterinary Laboratory Diagnosis, Faculty of Veterinary Medicine, Universiti Putra Malaysia, Serdang 43400, Selangor, Malaysia; 3Department of Biology, Faculty of Science, Universiti Putra Malaysia, Serdang 43400, Selangor, Malaysia; 4International Institute for Halal Research and Training (INHART), Level 3, KICT Building, International Islamic University Malaysia (IIUM), Gombak 53100, Selangor, Malaysia; hidayusaari@iium.edu.my; 5Department of Aquaculture, Faculty of Agriculture, Universiti Putra Malaysia, Serdang 43400, Selangor, Malaysia; gs51935@student.upm.edu.my

**Keywords:** vibriosis, streptococcosis, MAS, fish vaccination, oral vaccination

## Abstract

Multiple infections of several bacterial species are often observed under natural farm conditions. The infections would cause a much more significant loss compared to a single infectious agent. Vaccination is an essential strategy to prevent diseases in aquaculture, and oral vaccination has been proposed as a promising technique since it requires no handling of the fish and is easy to perform. This research attempts to develop and evaluate a potential feed-based polyvalent vaccine that can be used to treat multiple infections by *Vibrios* spp., *Streptococcus agalactiae*, and *Aeromonas hydrophila*, simultaneously. The oral polyvalent vaccine was prepared by mixing formalin-killed vaccine of *V. harveyi, S. agalactiae*, and *A. hydrophila* strains with commercial feed pellet, and palm oil as an adjuvant was added to improve their antigenicity. Thereafter, a vaccinated feed pellet was tested for feed quality analysis in terms of feed stability in water, proximate nutrient analysis, and palatability, safety, and growth performance using Asian seabass, *Lates calcarifer* as a fish host model. For immune response analysis, a total of 300 Asian seabass juveniles (15.8 ± 2.6 g) were divided into two groups in triplicate. Fish of group 1 were not vaccinated, while group 2 was vaccinated with the feed-based polyvalent vaccine. Vaccinations were carried out on days 0 and 14 with oral administration of the feed containing the bacterin at 5% body weight. Samples of serum for antibody and lysozyme study and the spleen and gut for gene expression analysis were collected at 7-day intervals for 6 weeks. Its efficacy in protecting fish was evaluated in aquarium challenge. Following vaccination by the polyvalent feed-based vaccine, IgM antibody levels showed a significant (*p* < 0.05) increase in serum against *Vibrio harveyi*, *Aeromonas hydrophila*, and *Streptococcus agalactiae* and reached the peak at week 3, 5, and 6, respectively. The high-stimulated antibody in the serum remained significantly higher than the control (*p* < 0.05) at the end of the 6 weeks vaccination trial. Not only that, but the serum lysozyme level was also increased significantly at week 4 (*p* < 0.05) as compared to the control treatment. The immune-related gene, dendritic cells, C3, Chemokine ligand 4 (CCL4), and major histocompatibility complex class I (MHC I) showed significantly higher expression (*p* < 0.05) after the fish were vaccinated with the oral vaccine. In the aquarium challenge, the vaccine provided a relative percentage survival of 75 ± 7.1%, 80 ± 0.0%, and 80 ± 0.0% after challenge with *V. harveyi*, *A. hydrophila*, and *S. agalactiae*, respectively. Combining our results demonstrate that the feed-based polyvalent vaccine could elicit significant innate and adaptive immunological responses, and this offers an opportunity for a comprehensive immunization against vibriosis, streptococcosis, and motile aeromonad septicemia in Asian seabass, *Lates calcarifer.* Nevertheless, this newly developed feed-based polyvalent vaccination can be a promising technique for effective and large-scale fish immunization in the aquaculture industry shortly.

## 1. Introduction

Aquaculture is the fastest-growing sector of agriculture in the world and accounts for almost 50% of the world’s food fish [1]. Globally, fish signifies 16.6% of animal protein sources and 6.5% of the total protein for human consumption [2]. With such a rapidly growing industry, the losses of any kind in production must be minimized. Although the fish culture is expanding worldwide, bacterial and viral infection incidence is also high throughout the globe [3,4]. Multiple infections of several bacterial species are often observed under natural farm conditions. The infections would cause a much more significant loss compared to a single infectious agent [5,6].

In Malaysia, *Vibrio* spp., *Streptococcus agalactiae*, and *Aeromonas hydrophila* are three bacterial pathogens that significantly affect the aquaculture industries [7,8,9]. They are the causative agents of vibriosis, streptococcosis, and motile aeromonad septicemia (MAS), respectively, in which the outbreak can pose a severe economic loss to any fish farming operation. Cultured fish such as Asian seabass, *Lates calcarifer*, Barramundi in Australia, and Siakap in Malaysia are vulnerable to diseases caused by *Aeromonas* sp., *Vibrio* sp., and *Streptococcus* sp. [10,11,12,13]. The body concentration of salt inside this fish is lower than the surrounding water; thus, it requires them to continuously drink water making the gastrointestinal tract over-exposed to waterborne infections [14]. Infected fish show common symptoms such as loss of appetite, hemorrhages, lethargy, and mortality [15,16,17].

The common practice in treating a bacterial fish disease involves the application of antibiotics. The drugs were widely used as a veterinary medicine for aquaculture treatments to control infectious bacterial disease, including streptococcosis, vibriosis, and MAS, to prevent any loss in aquaculture activities [18]. They are useful to cure bacterial diseases in the past before the extensive and frequent use of antibiotics in aquaculture nowadays resulted in resistance development in aquaculture pathogens, making them no longer effective in treating bacterial diseases at this moment [1]. This is due to some species of bacteria undergoing mutation in unfavorable conditions after using antibiotics to survive in a new condition [19]. Not only that, the usage of antibiotics increases the risk of occupational exposure of antibiotics to farmers, bioaccumulation and toxic actions on aquatic ecosystems, and antibiotic residue in cultured animals, which may be transferred and accumulated in human bodies [20]. Owing to the issues, the application of antibiotics is no longer encouraged. Therefore, vaccination is now considered the best approach to prevent specific disease outbreaks.

Vaccination is one of the alternatives that have been proposed to overcome disease-caused mortality and morbidity after the restriction of using antibiotics in aquaculture [21] due to vaccines being more effective and safer than antibiotics to humans and the environment [22]. As opposed to antibiotics that aim to kill or stop diseases, vaccines, on the other hand, stimulate the fish’s immune system for antibody production that protects the fish from diseases. Current vaccine applications for large-scale fish farming operations include inactivated, live-attenuated, and DNA vaccines.

Nowadays, numerous successful fish vaccination studies have been reported. For example, a study by Huang et al. [23] reported a DNA vaccine from a 1.02 kb DNA fragment inserted into pVAX1, which encodes a portion of the surface immunogenic protein (Sip) of *S. agalactiae*. The DNA vaccine provided more than 80% survival in Nile tilapia, *Oreochromis niloticus*, after bacterial challenge with *S. agalactiae*. Not only that, Nehlah et al. [24] developed a recombinant vaccine by cloning the OmpK and OmpW genes of *V. alginolyticus* into pET32 Ek/ LIC vector and expressed in *Escherichia coli*. The recombinant vaccine provided 100% protection against 10^9^ CFU mL^−1^ of live *V. alginolyticus* in juvenile hybrid groupers, *Epinephelus fuscoguttatus* × *Epinephelus lanceolatus*. Ghosh et al. [25] reported a live-attenuated fish vaccine against *Aeromonas hydrophila* in common carp, *Cyprinus carpio.* In their research, a live-attenuated strain of *A. hydrophila*, XX1LA, was screened from the pathogenic strain of *A. hydrophila* XX1 cultured on an antibiotic rifampicin-containing medium used as a live-attenuated vaccine candidate. The live-attenuated vaccine offered safety for up to 83.7% of survival after a bacterial challenge with wild-type *A. hydrophila*. Another study with successful fish immunization is a study by Wei et al. [26], which developed a *V. harveyi* formalin-killed cells vaccine against *V. harveyi* in hybrid grouper *Epinephelus fuscoguttatus × E. lanceolatus*. The vaccine provided more than 90% protection against the bacteria. Further studies are being conducted on fish vaccine research and development, hoping to prevent the diseases in an affordable, efficient, and safe way [27].

Vaccination is the most operative strategy to control infectious diseases in species with adaptive immunity as vaccination typically induces specific memory cells, which can mediate a fast-anamnestic response upon infection by the targeted pathogen [28]. Apparently, standard fish vaccination against streptococcosis, MAS, and vibriosis have been developed [29,30]. Most of the vaccines, however, were targeted at the single infection of respective pathogens. Developing a polyvalent vaccine that can be used to treat multiple infections by *Streptococcus agalactiae*, *Aeromonas hydrophila*, and *Vibrio* species will simultaneously provide practical ease of application, decreasing workload in comparison to the other ways of vaccination. Economically, countering multiple diseases using one application is cost-effective rather than buying a separate vaccine.

There are many ways of fish vaccination in which oral administration of the vaccine is one of them. Fish feed vaccination offers free-of-stress administration other than easy administration for a large number of fish [31], which is highly suitable for mass aquaculture activity. Oral vaccines are produced by either top coating the feed with antigen or mixing antigens into the feed during production. Delivery of antigen in or on fish feed offers the advantages of being stress-free and easy to administer to large numbers of fish at one time. In a previous study, we successfully developed an oral vaccine for vibriosis in marine fish, in which we included palm oil as an emulsion with a whole-cell vaccine containing formalin-inactivated *Vibrio harveyi* strain VH1. The vaccine mixture was processed into a commercial pellet as an alternative to the laborious injection and immersion delivery methods. We have successfully filed for a patent of composition and method for our feed-based vaccine against vibriosis (patent No.: PI2021000105).

The effort of developing a bivalent and polyvalent vaccine via oral administration is novel and is ongoing. From different points of view, polyvalent vaccination via feeding is advantageous. In the fish breeder’s eyes, oral vaccination provides practical ease of application while decreasing workload compared to the other ways of vaccination. Economically, countering multiple diseases using one application, in this case, polyvalent vaccine-infused feed, is cost-effective rather than buying a separate vaccine. To our knowledge, no report has been claimed to construct a polyvalent vaccine that can be used to treat multiple infections by *Streptococcus agalactiae*, *Aeromonas hydrophila*, and *Vibrio* species simultaneously. Therefore, this research study proposes a newly developed, oral feed polyvalent vaccine. This will help farmers treat either single or multiple infections, leading to streptococcosis, MAS, and vibriosis diseases simultaneously. This vaccinated feed will offer adequate protection for the fish against those diseases, is easy to handle, is safe for the animal and the environment, and is very cost-effective.

## 2. Materials and Methods

### 2.1. Bacterial Strains and Culture Conditions

The genotype of bacterial strains used in this study was described in detail in Table 1. The pathogenic *Vibrio harveyi* strain Vh1, used in this study, was previously isolated from diseased grouper from a local farm in Malaysia. This pathogenic strain was grown and maintained in selective thiosulphate-citrate-bile-salts-sucrose (TCBS) agar (Oxoid, Hampshire, England) followed by inoculation and incubation in tryptone soy broth (TSB) (Oxoid, Hampshire, England), with the addition of NaCl (1.5% w/v) for 24 h at 30 °C with 150 rpm. The cultures were stored by preserving in 20% glycerol at −80 °C. The cultures were retrieved on tryptic soy agar (TSA, Oxoid, Hampshire, England) + 1.5% NaCl plates incubated for 24 h at 30 °C. The pathogenic *A. hydrophila* strain Ah1sa5 used in this study was isolated from freshwater fish from local farms in Malaysia. The pathogenic strains were grown and maintained in selective *Aeromonas* agar base (RYAN) (Oxoid, Hampshire, England) for 24 h at 30 °C followed by inoculation and incubation in tryptone soy broth (TSB) for 24 h at 30 °C with 150 rpm. The cultures were stored by preserving in 20% glycerol at −80 °C. The cultures were retrieved on TSA plates incubated for 24 h at 30 °C. The pathogenic *S. agalactiae* strain Sa2k used in this study was originated from the outbreaks of streptococcosis in red tilapia kept in the cage-culture system at Kuala Lipis, Pahang, Malaysia, in 2007, where thousands of mortality occurred [30]. The pathogenic strains were grown and maintained in selective blood agar (Oxoid, Hampshire, England for 24 h at 30 °C followed by inoculation and incubation in tryptic soy broth (TSB, Oxoid, Hampshire, England) for 24 h at 30 °C with 150 rpm. The cultures were stored by preserving in 20% glycerol at −80 °C. The cultures were retrieved on TSA plates incubated for 24 h at 30 °C.

### 2.2. Reviving the Virulence and Preparation of Pathogenic Bacterial Strains

The pathogenic bacterial strains revived their virulency following the protocol of Koch’s postulate. A total of 200 μL of culture broth of the bacterial strains was administered by intraperitoneal (IP) injection into a healthy juvenile Asian seabass, *Lates calcarifer.* The infected fish was found dead within 24 h post-infection before samples of kidney, spleen, and liver were collected for bacterial isolation. The bacterial strain was successfully recovered from the kidney, spleen, and liver of dead fish using a sterile swab and subculture into selective agars at 30 °C with shaking at 150 rpm for 16 h. Single colonies of bacteria proceeded with PCR for identification. The use of Asian seabass in this study was approved by Animal Care and Use Committee Universiti Putra Malaysia (UPM/IACUC/AUP-R078/2019).

#### PCR Amplification Using 16S rRNA

According to the manufacturer’s protocol, pure colonies’ genetic DNA was extracted using the DNeasy Blood and Tissue kit (Qiagen). The genomic DNA was subjected to PCR amplification using 16S rRNA primers (Table 2). The PCR reactions were performed using REDiant 2× PCR Master Mix (FirstBase, Malaysia) in a final volume of 25 μL containing 2× PCR master mix, 1 µM of each primer, and 100 ng of template DNA. The gyrB cycle condition was an initial denaturation at 95 °C for 5 s, followed by 33 cycles of 95 °C for 1 min; 59 °C for 2 min 15 s and 72 °C for 1 min 15 s, and a final extension of 72 °C for 10 s. The amplification was performed in an Eppendorf Mastercycler Nexus Thermal Cycler (Eppendorf, Hamburg, Germany). Direct sequencing of purified PCR products was performed by First Base (Malaysia).

### 2.3. Feed-Based Vaccine Preparation

In this study, we used a specific strain, *V. harveyi* strain Vh1, from our local isolates that had been shown to induce strong antigenic responses on its homologous OMPs antigen and cross-reacted against heterologous OMPs antigens of *V. parahaemolyticus* strain VPK1, *V. alginolyticus* strain VA2, and *Photobacterium damselae* strain PDS1 at a molecular weight of 32 kDA. The strains were later prepared as an inactivated vaccine using 0.5% formalin and were tested for an in-vivo challenge using *Artemia* sp as a host model. The inactivated vaccine was found to provide protection up to 83.3% against wild-type *V. harveyi* [35]. A similar strain was used in Aslizah et al. [32], Chin et al. [36], and Chin et al. [37], and showed significant protection against *Vibrio* spp. In comparison, *Streptococcus agalactiae* strain Sa2k used in this study was previously used as an antigen in a feed-based vaccine against streptococcosis in red tilapia, *Oreochromis niloticus* and was found to provide 100% survival against *S. agalactiae* [30]. Nevertheless, *A. hydrophila* strain Ah1sa5 used in this study was reported to be used as an antigen in a recombinant cells vaccine and was found to provide 100% survival against wild-type *A. hydrophila* in African catfish, *Clarias gariepinus* [33].

*Vibrio harveyi* strain Vh1, *Streptococcus agalactiae* strain Sa2k, and *Aeromonas hydrophila* strain Ah1sa5 were cultured on selective agars incubated at 30 °C for 24 h. The bacterial concentration was determined using the standard plate count technique and adjusted to 6.7 × 10^7^ CFU/mL. According to a method previously described by Ismail et al. (2017a), the bacteria were formalin-killed with modifications. Following inactivation, the suspension was then streaked onto selective agar and incubated for 24 h at 30 °C to ensure that all respective bacteria were killed and there was no contamination. The individual bacterial suspension was mixed equally, and palm oil was added to a final concentration of 10% before the mixture was thoroughly mixed with pelleted feed to provide a final concentration of 10^6^ CFU/g of feed. For control, only phosphate buffer saline (PBS) and palm oil were mixed with the pelleted feed.

### 2.4. Feed Quality Analysis

Feed quality analysis was conducted in this study to determine the vaccinated pellet quality in terms of nutrient composition, physical stability in water, fish acceptability, and growth performance after incorporated with the inactivated whole-cell vaccine by comparing it with an available commercial pellet (Star Feed, Star Feed Mills SDN. BHD, Malaysia).

#### 2.4.1. Proximate Analysis

The feed used in the experiments were analyzed for proximate composition. The crude protein, lipid, carbohydrate, fiber, ash, and moisture content were determined according to the AOAC (1997) methods and by referring to Sulaiman et al. [38]. Crude lipid was determined by solvent extraction with petroleum ether (Foss Tecator Lipid Analyzer), while crude protein percentage was measured by a protein analyzer (Foss 2400 Kjeltec Analyzer Unit). Dry matter was evaluated by oven drying the samples at 105 °C to a constant weight, and total ash content was determined by incinerating samples at 600 °C for 5 h. Finally, crude fiber was estimated by acid digestion followed by alkaline digestion (Foss Fibertec 2010, Hot Extractor System).

#### 2.4.2. Physical Stability of Feed in Water

The stability of the reformulated pellet with an inactivated whole-cell vaccine in water was tested by the static water method according to Obaldo et al. [39]. A total of 2 g of each pellet (commercial and vaccinated) in triplicate was leached for desired immersion time (0, 1, 2, 3, 4, 5, 6, and 7 h) in a 250 mL flask filled with 100 mL of water. After the desired leaching time, the immersed pellet was filtered using Whatman filter paper no. 1. The recovered and original feed sample was dried in the oven at 100 °C for 24 h. The recovered solid was analyzed for pellet stability in terms of dry matter retention using the following formula: [Feed remaining (g)/initial feed (g)] × 100(1)

#### 2.4.3. Palatability Test

The acceptability of the fish to the feed was determined, according to Dong et al. [40]. In this study, we used Asian seabass, *Lates calcarifer* (15 ± 2.6 g body weight) as a model for fish acceptability to the feed. A total of 30 Asian seabass were divided into 2 groups in triplicate. Group 1 was fed with commercial feed while group 2 with vaccinated feed. Fish were starved for 24 h before feeding. Fish from each tank were fed with 2 g pellets, and the uneaten were removed and counted 1 h later. The ingestion ratio was used to evaluate the palatability using the formula:R_i_ = weight of pellet ingested (g)/weight of pellets fed (g)(2)

#### 2.4.4. Growth Performance

A new batch of fifty Asian seabass (*n* = 50, 2.31 ± 0.08 g body weight) was divided equally into two treatments and cultured individually in 30 L glass tanks. Fish were fed with either vaccinated or unvaccinated (control) pellets to satiation two times daily and weighed weekly during the 6-week test. Growth response and nutrient use parameters included specific growth rate (SGR) and feed conversion ratio (FCR) and were calculated using the indicator shown below:Total weight gain (WG) = Final body weight (g) − Initial body weight (g)(3)
Specific growth rate (SGR) = 100 × [((Ln Final body weight) − (Ln Initial body weight))/duration] (4)
Feed conversion ratio (FCR) = Feed intake (g)/weight gain (g)(5)

### 2.5. Vaccination Trial

#### 2.5.1. The Fish

A total of 300 (*n* = 300) Asian seabass with an average body weight of 15.8 ± 2.6 g were purchased from a commercial fish farm in Melaka, Malaysia. The fish were acclimatized for seven days in a 500 L fiber-glass tank installed with a recirculating water system and filled with pre-treated (sand filter and UV light) seawater. The following water parameters were measured using a YSI Pro Plus multiparameter instrument (Yellow Springs Instrument) and maintained throughout the acclimatization and experimental periods: temperature at 25.28 ± 0.81 °C, pH at 7.66 ± 0.06, salinity at 27.22 ± 0.78 ppt, and dissolved oxygen at 5.93 ± 0.25 mg/L. The fish were fed with a commercial fish pellet twice a day ad libitum. Before the experiment, 30 fish were randomly selected and sampled to examine bacterial recovery from blood, kidney, liver, and spleen to ensure the fish’s health status. The fish were found to be healthy, and no bacteria could be detected in all the fish examined.

#### 2.5.2. Experimental Design

The vaccination trial was carried out for a period of six weeks. A brief experimental design and feeding regime are shown in Figure 1. At the start of the trial, the fish were divided into two groups in triplicate. The fish were not fed for a day before the experiment’s start to ensure maximum feed-vaccine uptake. Group 1 was given the control feed that contained PBS, while group 2 was fed with the feed-based vaccine for 5 consecutive days following Mao et al. [41], Samuelsen [42], Monir et al. [43], Sun et al. [44], Sun et al. [45], Tu et al. [46], and, Wang et al. [47] on weeks 0 and 2. The fish were fed with the vaccinated feed at 5% body weight. The feeding and husbandry practices of the farm were maintained.

Serum samples were collected weekly from six fish of each group for antibody and lysozyme study and the gut for gene expression analysis for a period of six weeks. At the same time, water quality parameters such as pH, temperature, salinity, dissolved oxygen, and ammonia-nitrogen were monitored using YSI Professional Plus (Yellow Spring Instrument, Yellow Spring, OH, USA) and ammonia-nitrogen spectrophotometer (HACH Company, Loveland, CO, USA) until the end of the 6-week experimental period. Fish mortalities and abnormal features were recorded, while the survival rate was calculated at the end of the 6-week study. Vaccine safety was determined along the vaccinated period to ensure any adverse effects of vaccinated feed on the fish.

#### 2.5.3. Challenge Test

The efficacy of vaccines in fish was tested by experimental challenge with the respective bacterial pathogens at day 28 post-immunization. The challenge test was carried out in duplicate in a 150 L freshwater tank system, which contained 10 fish each. Both vaccinated and unvaccinated fish were challenged with either *V. harveyi*, *S. agalactiae*, or *A. hydrophila* by intraperitoneal injection with 10^7^ CFU bacteria/fish. Mortality was recorded for 14 days. The protective efficacy of vaccination was evaluated by calculating cumulative percent mortality (CPM) of each treatment and relative percent survival (RPS) of the vaccinated groups for 7 days after the challenge using the following formula:Cumulative percent mortality (CPM) = (the number of fish mortality/the total number of fish) × 100(6)
Relative percentage survival (RPS) = [(1 − (average CPM in the vaccinated group/average CPM in the control group)) × 100](7)

### 2.6. Sample Processing

#### 2.6.1. Enzyme-Linked Immunosorbent Assay (ELISA)

Serum samples were subjected to indirect ELISA to determine the IgM level, according to Firdaus-Nawi et al. [30], with minor modifications. Flat-bottom microtitre plates were coated with 100 µL coating antigen containing 10^5^ CFU/mL of formalin-killed whole cells of *V. harveyi*, *S. agalactiae*, and *A. hydrophila* separately in carbonate-bicarbonate buffer per well. The plates were left overnight at 4 °C prior to two times washing with phosphate buffer saline +0.05% Tween 20 (PBST). Plates were then blocked with 200 µL of 1% bovine serum albumin (BSA) diluted in PBS and incubated for 1 h at 37 °C. Next, after the reaction well was washed three times with PBST, 100 µL of 1:1000 serum, mucus, and gut lavage diluted in PBS were inserted into the reaction well and incubated again for 1 h at 37 °C. Unbounded antibodies were removed by washing the reaction well thrice with PBST. Specific IgM was detected using anti-Asian seabass IgM monoclonal antibody (Aquatic Diagnostics Ltd., Stirling, UK, 1/33 in PBS, 1 h) followed by incubation with anti-mouse HRP (1/5000, Nordic, 1 h). After three times washed with PBST, 100 µL of TMB substrate solution (Thermo Fisher Scientific, Waltham, MA, USA) was added to the reaction well to detect the bound conjugate before the reaction was stopped with 0.2 mol/L sulphuric acids. Values were obtained by measuring the absorbance at 450 nm using a Multiskan spectrum microplate reader (Thermo Fisher Scientific Inc., Madison, WI, USA). The cut-off value was the highest possible true-positive rate that was used as an indication of protection. It was determined by performing ELISA on 100 samples collected from non-immunized and uninfected fish [48].

#### 2.6.2. Serum Lysozyme Activity

Serum lysozyme activity was measured with turbidimetric assays based on Byadgi et al. [49]. A total of 100 μL of samples were put into wells of a microplate, then we added 100 μL of a substrate (0.2 mg *Micrococcus lysodeikticus* lyophilized cell, Sigma, Columbia, MO, USA)/mL PBS, pH 7.4. Their absorbance was measured after 1 and 60 min, which were incubated with gentle shaking at constant room temperature, at 450 nm with a Multiskan spectrum microplate reader (Thermo Fisher Scientific Inc., Madison, WI, USA). A unit of enzyme activity was defined as the amount of enzyme that causes a decrease in absorbance of 0.001/min.

#### 2.6.3. Quantitative Real-Time PCR (qPCR)

Quantitative real-time PCR (qPCR) was performed using SYBR Green real-time PCR mix (QuantiNova™, Qiagen™, Hilden, Germany) following the manufacturer’s instructions. For each sample, 3 µL of cDNA sample was added into 10 µL of 2X SYBR Green PCR master mix, 1.4 µL of each forward and reverse primers of immune-related genes (Table 3), and 1.2 µL of nuclease-free water into a PCR tube. Each cDNA sample group was prepared in triplicate. The samples were run in the Rotor-Gene Q real-time PCR machine (Qiagen™, Hilden, Germany). The resulting threshold cycle (CT) values were analyzed and recorded using Rotor-Gene Q software (Qiagen™, Hilden, Germany). The qPCR program was as follows: initial denaturation at 95 °C for 10 min, followed by 40 cycles of denaturation at 95 °C for 15 s, and annealing/elongation at 60 °C for 60 s. A melting curve was run for each primer set to confirm the reaction specificity. The relative expression of each immune-relative gene was determined by comparing it with the expression level of the β-actin gene as the housekeeping gene using the Livak method, 2−ΔΔCT method.

### 2.7. Statistical Analysis

*t*-test was used to determine the significant differences between vaccinated and control groups (SPSS 22.0 package, SPSS Inc., Chicago, IL, USA). The results were considered significant at *p* < 0.05.

## 3. Results

### 3.1. Feed Quality Analysis and Growth Performance

The proximate compositions between vaccinated and control feed showed no significant difference (*p* > 0.05; Table 4) except for the moisture content. The vaccinated feed showed significantly higher moisture content than the commercial feed (*p* < 0.05). Figure 2 shows the comparison of pellet water stability results for commercial and formulated vaccinated feed groups. There were no significant differences between commercial formulated vaccinated feed and commercial feed during the 7 h immersion in the water (*p* > 0.05; Figure 2).

According to previous studies by Dong et al. [40], the palatability results were valued with the ingestion ratio (R_i_). The vaccinated feed showed significantly better palatability than commercial feed (*p* > 0.05; Figure 3), and their R_i_s was nearly 2. The growth performance of Asian seabass is presented in Figure 4 and Table 5. Throughout the experiment, fish appeared to be healthy, and no mortality occurred. After six weeks of observation, post-vaccination with the commercial feed with vaccine indicated that the growth parameters, including the growth performance, SGR, and FCR, showed significantly improved growth parameters (*p* < 0.05; Table 5) compared to the commercial feed.

### 3.2. IgM Antibody Responses

ELISA results indicated that vaccinated Asian seabass produced significantly higher antibody titers than the control (*p* < 0.05; Figure 5, Figure 6 and Figure 7). Nevertheless, the rate of responses was different depending on the coating antigens. When the serum was tested against *V. harveyi*, the IgM antibody titers resulted significantly higher than the control (*p* < 0.05; Figure 5) as early as week 2. The IgM antibody levels were higher than the normal cut-off value until week 6. Meanwhile, for the IgM antibody titers against *S. agalactiae*, the value only started to show significantly higher titers than the control (*p* < 0.05; Figure 6) at week 4 to week 6. In contrast, the IgM antibody level against *A. hydrophila* started to show a significant increase compared to the control (*p* < 0.05; Figure 7) from week 1 until week 4 before slightly dropping in week 5. The antibody titers were later increased at week 6.

### 3.3. Lysozyme Activity

As shown in Figure 8, the vaccinated groups’ serum lysozyme activity increased every week after immunization, and these values were significantly higher than the PBS control group (*p* < 0.05) from 2 weeks post-immunization and peaked at week 6.

### 3.4. Gene Expression

The expression of immune-related genes of chemokine ligand 4 (CCL4), dendritic cells, C3, and MHC I after orally immunized with the polyvalent vaccine was determined (Figure 9). As shown in Figure 9, the relative expression of the MHC 1 transcript level was increased significantly compared to the control (*p* < 0.05) at week 3 and was maintained higher than the control (PBS; *p* < 0.05) afterward, during the entire trial. For immune genes of C3, dendritic cells, and CCL4, gene expression increased as early as week 1 for both genes and peaked at weeks 1, 3, and 6, respectively.

### 3.5. Efficacy of the Feed-Based Polyvalent Vaccine

The protective efficacy of the oral polyvalent vaccine via bacterial laboratory challenge was showed in Table 6. Negative control in the unchallenged groups achieved 100% survival. However, the unvaccinated fish in the challenged group were infected with either *V. harveyi*, *S. agalactiae*, or *A. hydrophila*, began to die on day 1, and hit 100% mortality after day 7 to day 10 of post-challenge (Figure 10). On the other hand, the oral vaccine significantly improved fish resistance to the pathogens. In fact, the relative percentage survival (RPS) of the vaccinated fish was 75 ± 7.07% for *V. harveyi* challenge, 80 ± 0.00% for *S. agalactiae* challenge, and 80 ± 0.00% for *A. hydrophila* challenge. Overall, the feed-based polyvalent vaccine proved to be highly beneficial since RPS of the vaccinated group was 75% to 80% in fish following primary immunization at Week 0 and booster vaccinations at Week 2.

## 4. Discussion

In the aquaculture industry, the usage of antibiotics and chemotherapeutic agents such as lime, saponin, and formalin were reported to provide positive outcomes [52,53]. Unfortunately, in intensive culture, the usage of antibiotics and chemotherapeutics can lead to detrimental consequences such as bioaccumulation, pollution, and antibiotic resistance that can be transmitted to wild and human pathogenic microbes, therefore posing a hazard to human health and also impacting socio-political and environmental problems as demonstrated in some countries [53]. Therefore, developing a new measure to control the disease is crucial to the aquaculture industries. Rather than therapeutics to treat the diseases and their symptoms, vaccines are distinctive modern medications that provide adequate protection against the onset and development of specific infectious diseases [22].

Many experimental vaccines have been introduced to protect fishes against acute bacterial infections in recent years, including furunculosis, vibriosis, yersiniosis, streptococcosis, and edwardsiellosis [54,55,56,57]. Most aquaculture vaccines developed for fish farmed for human consumption must be safe for the animal and the consumer, cheap, easy to use, and can stimulate long-term immunity against target pathogens [55]. Thus, in this current study, we developed a polyvalent, inactivated feed-based vaccine that may potentially fulfill all of the requirements. Polyvalent vaccines are especially crucial in aquaculture due to various bacterial species in the breeding environment [57], and economically, countering multiple diseases using one application is cost-effective rather than buying separate vaccines.

Inactivated vaccines showed superior protection efficiency among the various vaccine types compared with live-attenuated, recombinant, and DNA vaccines [58,59]. The inactivated vaccine is highly beneficial as it stimulates the antibody-related portion of the immune responses, requiring minimal preparation cost, high stability, and safety [60]. There are many ways of fish vaccination in which oral administration of the vaccine is one of them. Fish feed vaccination offers free-of-stress administration other than easy administration for a large number of fish [31], which is highly suitable for mass aquaculture activity. This study also highlights three significant pathogens in Malaysia: *Vibrio* spp., *Streptococcus agalactiae*, and *Aeromonas hydrophila* that affect many aquaculture industries in this country. They are the causative agents of vibriosis, streptococcosis, and motile aeromonas septicemia (MAS), respectively, in which the outbreak can pose a severe economic loss to any fish farming operation. Thus, this current study used the local isolates of *Vibrio* spp., *Streptococcus agalactiae*, and *Aeromonas hydrophila* from the infected farm and included the three pathogens in the vaccine preparation.

The purpose of the present study was to develop and evaluate a locally prepared feed-based polyvalent vaccine that protects aquaculture fishes against common infections caused by the three pathogens, *Vibrio* spp., *Streptococcus agalactiae*, and *Aeromonas hydrophila*, and study the immune response in Asian seabass, *Lates calcarifer*, as a fish host model.

Good nutrition in animal production systems is essential to the economical production of a healthy, high-quality product. In fish farming (aquaculture), nutrition is critical because feed typically represents approximately 50 percent of the variable production cost. Fish nutrition has advanced dramatically in recent years by developing new, balanced commercial diets that promote optimal fish growth and health. The development of new species-specific diet formulations supports the aquaculture industry as it expands to satisfy the increasing demand for affordable, high-quality fish and seafood products. Not only that, in fish farming, a safe and effective vaccine by the oral route is highly needed [23]. In this study, the commercially available feed was used and combined with the polyvalent vaccine. This is to ensure fish already tolerate and are adapted to the feed; thus, the feed will be easily accepted by the fish. The problem with the commercial feed’s reformulated feed is that some nutrient composition may be changed during the process, affecting the required nutrient absorbed by the fish. In this study, nutrient compositions such as protein, lipid, carbohydrate, and ash are not significantly changed. However, moisture was significantly higher than the commercial pellet. The moisture may be increased due to additional moisture included when the vaccine was added to the ground pellet before being re-pelleted again using an extruder. Although the moisture was significantly higher than the commercial pellet, it is still under an acceptable range. Too much moisture in the feed pellet may lead to the feed pellet becoming moldy and shortening the feed’s shelf life [61].

Pellet water stability is one of the important quality parameters in the manufacture of aquaculture diets. Pellet disintegration is more important in benthic fish such as sea bass and grouper, making sinking and water-stable feed necessary. Pellets with low water stability may disintegrate easily, the nutrients leached in the water before fish intake, and may deteriorate the cultured water quality [39]. In this study, the feed was reformulated and reprocessed to incorporate the vaccine into the feed, affecting the feed’s stability in water. However, results showed that the vaccinated feed had high water stability as the pellet was still intact after 7 h of immersion in water. No significant difference was found in the water stability between the vaccinated pellets and the commercial pellets, showing that the vaccinated pellet is stable in water, similar to the commercial pellet.

Palatability or acceptability of the feed to the fish is crucial. Feed with low palatability may reduce the fish’s feed intake, thus making the necessary nutrient or antigen, in this case, failed to be delivered inside the gut. However, in this study, the vaccinated feed has high palatability due to the palm oil adjuvant. A study by Ayisi et al. [62] reported that feed included with palm oil enhances the fish’s feed intake. High palatability leads to increased feed consumption, improving overall feed and nutrient intakes [63]. On the other hand, low feed palatability plays an important role in feed intake reduction and fish growth retardation [40].

Furthermore, the vaccinated fish show higher growth performance than the control group. These results were dissimilar with Fraser et al. [64], in which the authors reported that vaccination would reduce the growth of fish due to an increase in regular metabolic rate due to continuous stimulation of the immune system. However, Chalmers et al. [65] and Reyes et al. [66] suggested that oral vaccine did not affect normal fish growth, suggesting any detrimental effect over nutrient assimilation. On the other hand, in this current study, Beck et al. [67] and Ismail et al. [5] exhibited significantly enhanced growth performance after oral vaccination. The growth enhancement in this study can be attributed to the usage of palm oil as an adjuvant. Palm oil was reported to be high in saturated fatty acids (about 50%) [68], thus, providing adequate energy for fish growth [62].

The current study showed that the serum IgM antibody levels were significantly increased after immunization with the polyvalent vaccine. This may be due to the fact that the antigens from the polyvalent vaccine managed to be transported and reached the end gut segment of fish and induced both local and systemic immune responses [69]. Firdaus-nawi et al. [30] reported that the second gut segment of fish plays a significant role in oral vaccination as it participates in the transportation and presentation of antigens from the intestinal lumen to intra-epithelial macrophages. These macrophages ingest the antigen presented by this route and re-migrate from the intestine to lymphoid organs, mainly the spleen and kidney, before initiating a systemic immune response [70]. B cells are activated when the immune system is stimulated through the epitope’s specific binding to the B cell receptor. This binding event sends a signal to the nucleus through accessory proteins that trigger B cell receptors’ crosslinking to the co-stimulatory signal [71]. Once activated, B cells proliferate and mature to secrete specific immunoglobulins such as IgM against the detected epitope [72]. These results were in line with many studies such as Adelmann et al. [73], Firdaus-Nawi et al. [30], Chideroli et al. [74], Ismail et al. [5], and many more in which IgM antibody was significantly increased after oral vaccination.

The vaccinated feed also significantly improved the lysozyme activity in fish after seven days of vaccination until the six-week vaccination period. Lysozyme is involved in innate immunity in fish and plays a crucial role in mediating protection against bacterial invasion in the presence of complement [75]. On the other hand, the complement system plays a critical role in alerting and clearing potential pathogens in the host, according to Secombes and Wang [76]. This result is supported by the significantly higher expression of complement component 3 or C3, which peaked at week 1 and was significantly higher than the control until the end of the vaccination period. Other immune-related genes also showed significantly higher expression than the control. Chemokine ligand 4 (CCL4) and major histocompatibility complex class I (MHC I) were high in the gut after vaccination. The CCL4 is one of the chemo-attractant cytokines that regulate the relocation of immune cells in tissue and is crucial for innate immunity [36], and the MHC I molecules are crucial for representing peptide antigens on antigen-presenting cells of innate immunity to CD8 + T-lymphocytes (cytotoxic T cell) of adaptive immunity in teleost fish [77,78]. The higher expression of CCL4 in the gut following oral vaccination with the feed-based whole-cell polyvalent vaccine could be related to the high population of lymphocytes in the gut after oral vaccination [30]. Meanwhile, high expression of MHC I in the gut may be related to the high expression of CCL4 as a study by Sun et al. [79], which reported that high expression of CCL4 induces up-regulation of MHC I in silverfish, *Trachinotus ovatus.* Other than that, it may be because the abundance of T- lymphocytes (T cells) after oral vaccination with the feed-based polyvalent as T cells are abundant in teleost mucosal tissue [36]. Finally, the dendritic cell, a type of antigen-presenting cells (APCs) that act as a bridge that connects innate immune recognition and specific immune memory, is also highly expressed in this study. APCs are crucial in adaptive immunity as they help to recognize antigens, which later signal antibody responses and pathogen elimination by primed B- and T-lymphocytes [50]. The dendritic cells were significantly high since week 1, and peaked at week 3, which may be due to the inclusion of the bacterial antigens inside the fish’s feed. The gene expression results are similar to other research on fish vaccinations, such as a study by Sun et al. [44], which reported vaccination with VhhP2 induced the expression of several immune-related genes, especially those encoding MHC. Another study by Chin et al. [36] on the efficacy of bath vaccination with a live-attenuated *Vibrio harveyi* against vibriosis in Asian seabass showed higher expressions of the CCL4 and MHC I genes than the control after 12 h vaccination. Similar results with C3 gene expression in Bao et al. [80] were reported to be highly expressed in turbots after vaccination with combined live *Vibrio anguillarum* and *Edwardsiella piscicida* vaccines. This study was in line with Zoccola et al. [50], in which the authors reported higher expression of dendritic cells after antigen introduction in Asian seabass.

The newly developed feed-based vaccine is a good vaccine, as the vaccination trial in this current study resulted in a high survival rate, with 75 to 80% protection. Chettri et al. [81] concluded that a good vaccine is a vaccine that can provide more than 70% protection. Similarly, Gravningenet al. [82] reported 80% survival, Romer-Villumsen et al. [83] reported 78% survival, and Fredriksen et al. [84] reported 77.5% survival following fish vaccination. The improved survival after vaccination is beneficial to farmers as the harvest increases.

This feed-based vaccine comprises a combination of formalin-killed of the most important pathogenic bacteria that cause considerable losses in the aquaculture industry; thus, the results in this study are in line with similar with recent researches, which reported that a mixture of protein types and antigen sites as a bivalent or multivalent vaccine could improve the immune protection in fish [85,86].

Although, monovalent vaccines using a single bacterial antigen are well known to combat diseases in fish. However, vaccination route such as injection requires the fish be vaccinated more than one time as multiple infections may occur in a farm. This will cause handling stress to the fish and increase the vaccination costs. Thus, a safer approach is to mix monovalent vaccines into bivalent formulations to mitigate the problems. Not only that, nowadays, the usage of polyvalent or multivalent adjuvanted vaccines (4 to 6 antigens in combination) is being introduced to the farmers and has significantly improved the vaccination strategy. Thus, this strategy can reduce a considerable amount of money and yield higher fish production [87]. A study by Liu et al. [88] reported that the bivalent DNA vaccine induced more robust immune responses and was more protective than the monovalent DNA vaccine in the Japanese flounder (*Paralichthys olivaceus*) model. In other studies, rainbow trout and Atlantic salmon (*Salmo salar*) presented higher stimulation of the immune system after administering polyvalent vaccines, even better than the monovalent vaccines in the salmon case [89]. However, there are some disadvantages with the polyvalent vaccine, which was reported to provide short duration and low immunoprotection efficacy of inactivated vaccines [80]. Caution must be taken to formulate polyvalent vaccines because there may be an antigen competition issue, mainly when administered by injection [90]. A polyvalent vaccine’s effectiveness is regulated by individual antigen concentrations, cross-reactivity, and competition between different antigens [89].

It is unknown if the polyvalent feed-based vaccine can provide cross-protective efficacy against different bacteria strains across different fish species. Thus, more vaccination trials should be conducted against different species and strains of bacteria in marine and freshwater fish to address this issue. Other topics that should be considered in future studies include antigen dose, antigen administration duration, booster vaccination intervals, and protection immunity duration. Although these issues still need to be addressed, this study’s results illustrate that oral vaccination with the feed-based polyvalent vaccine is a promising method for a comprehensive immunization regime.

## 5. Conclusions

Oral immunization using feed-based, inactivated, whole-cell vaccine is a viable option against multiple aquaculture diseases, especially endemic vibriosis, as it provides intense immune stimulation that protects the fish, reduces infection rate, and improves the growth performance of the fish. Subsequently, survival could be improved.

## Figures and Tables

**Figure 1 vaccines-09-00368-f001:**
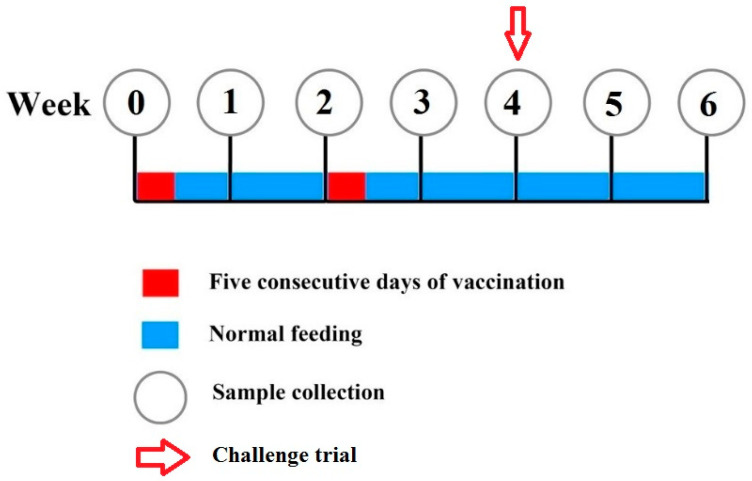
Feeding regime.

**Figure 2 vaccines-09-00368-f002:**
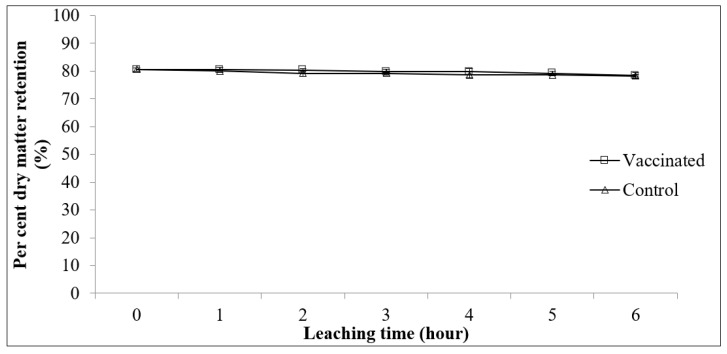
Percent dry matter retention (mean ± SE, *n* = 3) as a function of leaching time for commercial feed (control) and reformulated vaccinated feed using the static water method.

**Figure 3 vaccines-09-00368-f003:**
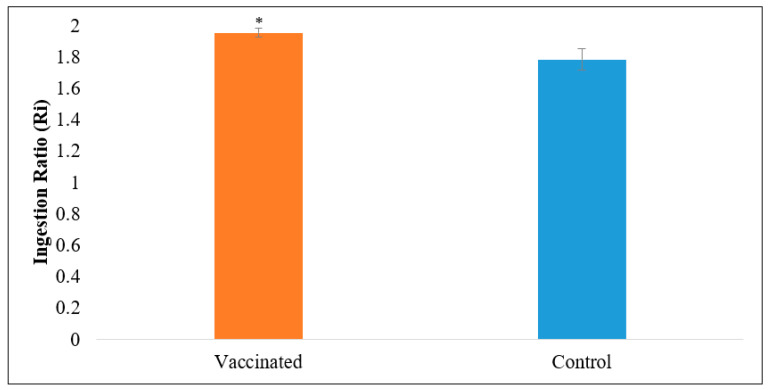
The ingestion ratio of vaccinated and commercial feed for Asian seabass, *Lates calcarifer* (mean ± SE, *n* = 15). Asterisks indicate a significant difference (*p* < 0.05) from the control.

**Figure 4 vaccines-09-00368-f004:**
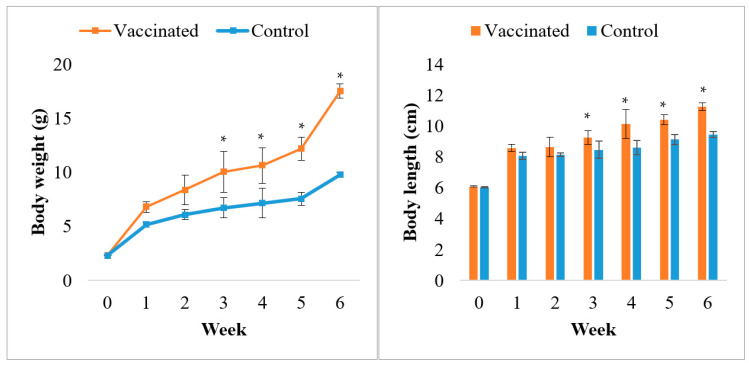
The fish growth performance based on total body weight and body length of the Asian seabass (mean ± SE, *n* = 25) during 6 weeks of vaccination trial period. Asterisks indicate a significant difference (*p* < 0.05) from the control.

**Figure 5 vaccines-09-00368-f005:**
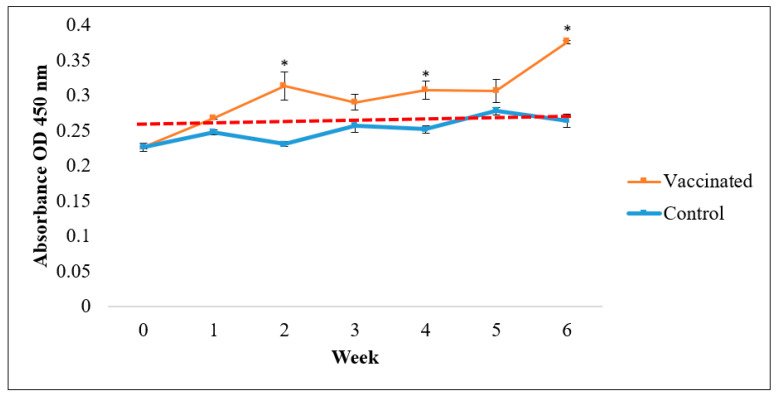
Specific antibody (IgM) levels against *V. harveyi* in vaccinated Asian seabass (mean ± SE, *n* = 6) 6 weeks post-vaccination. Asterisks indicate a significant difference (*p* < 0.05) from the control.

**Figure 6 vaccines-09-00368-f006:**
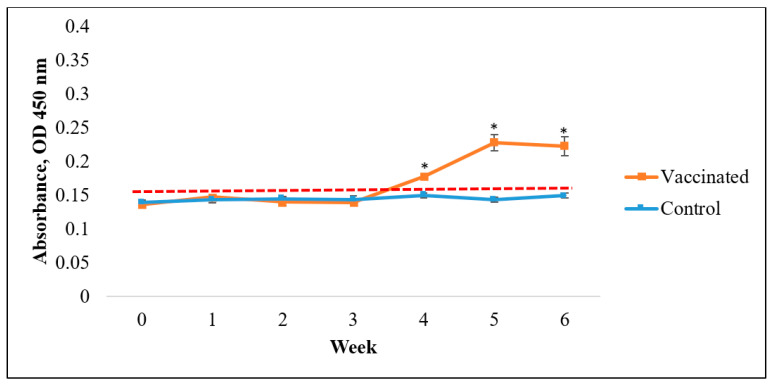
Specific antibody (IgM) levels against *S. agalactiae* in vaccinated Asian seabass (mean ± SE, *n* = 6) 6 weeks post-vaccination. Asterisks indicate a significant difference (*p* < 0.05) from the control.

**Figure 7 vaccines-09-00368-f007:**
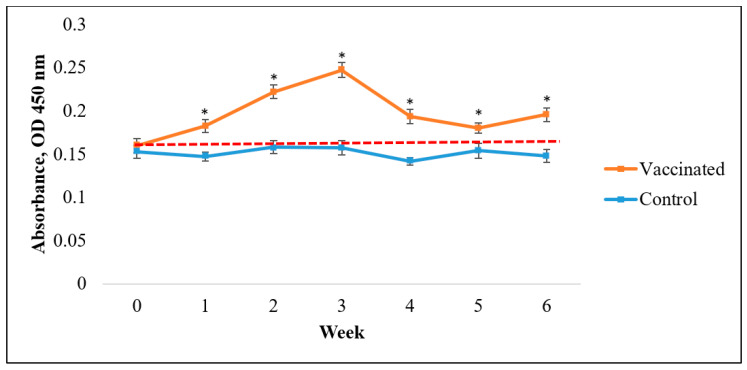
Specific antibody (IgM) levels against *A. hydrophila* in vaccinated Asian seabass (mean ± SE, *n* = 6) 6 weeks post-vaccination. Asterisks indicate a significant difference (*p* < 0.05) from the control.

**Figure 8 vaccines-09-00368-f008:**
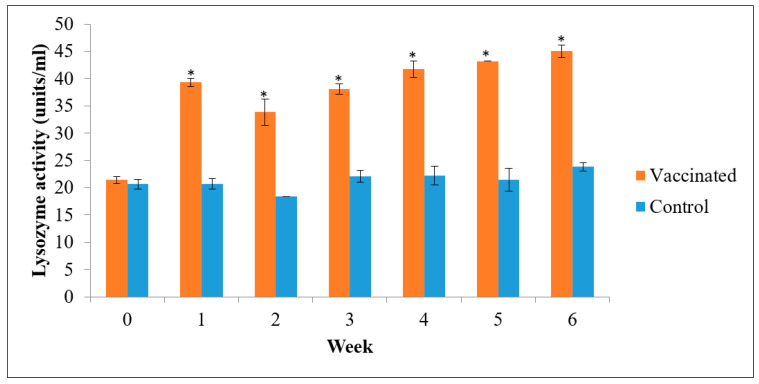
Lysozyme activity of *L. calcarifer* (mean ± SE, *n* = 6) vaccinated with oral feed-based polyvalent vaccine during 6 weeks vaccination trial. Asterisks indicate a significant difference (*p* < 0.05) from the control.

**Figure 9 vaccines-09-00368-f009:**
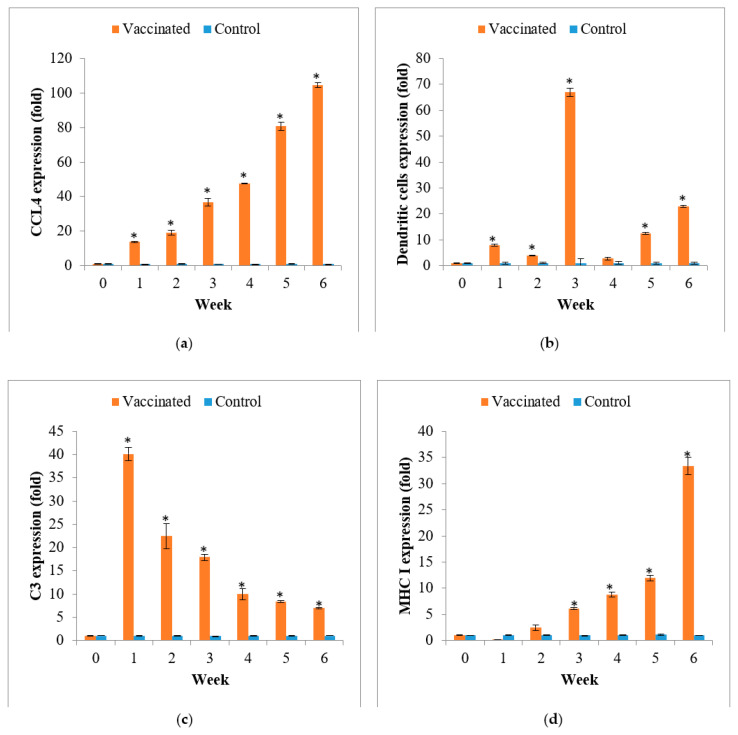
The relative expression of (**a**) CCL4, (**b**) dendritic cells, (**c**) C3, and (**d**) MHC I to β-actin in the gut after immunization in Asian seabass (mean ± SE, *n* = 6). Asterisks indicate a significant difference (*p* < 0.05) from the control.

**Figure 10 vaccines-09-00368-f010:**
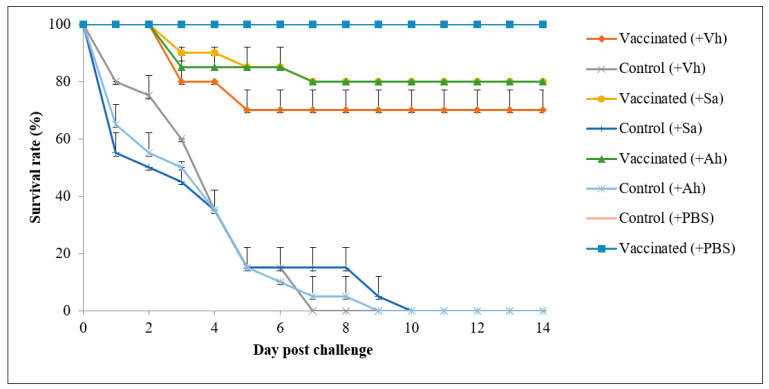
The survival rate of Asian seabass from vaccinated and non-vaccinated control groups after challenging with *V. harveyi* (+Vh), *S. agalactiae* (+Sa), *A. hydrophila* (+Ah), and PBS buffer (+PBS). Each treatment was performed in duplicate with 10 fish per tank. The upper half part of the standard deviation bars is shown.

**Table 1 vaccines-09-00368-t001:** Bacterial strains genotypes used in this study.

Bacterial Strain	Description/Genotype	Source	Ref.
Pathogenic *V. harveyi* strain Vh1	Wild type, local isolate	Lab collection	[32]
Pathogenic *A. hydrophila* strain Ah1sa5	Wild type, local isolate	Lab collection	[33]
Pathogenic *S. agalactiae* strain Sa2k	Wild type, local isolate	Lab collection	[30]

**Table 2 vaccines-09-00368-t002:** Primers used in PCR amplification of 16S rRNA.

Primers	Primer Sequence (5’-3’)	Tm (°C)	Expected Size (bp)	Reference
16S rRNA_R	AGAGTTTGATCCTGGCTCAG	54.0	1541	[34]
16S rRNA_F	GGTTACCTTGTTACGACTT			

**Table 3 vaccines-09-00368-t003:** Primers used in quantitative real-time PCR (RT-qPCR).

Target	Sequences (5’-3’)	Product Size	Annealing Temperature (˚C)	References	Note
Dendritic cells	F:AACAGCACACGCTCACTCAC R:CGATCATGTGAGCCTTGAGA	153	60	Zoccola et al. [50]	Initiate an adaptive immune response
C3	F:GCAATCCTCCACAACTACAG R:ACTCTGACCTCCTGACGATAC	93	59	Mohd-Shaharuddin et al. [51]	Innate defense against common pathogens
CCL4	F:TCCTCGTCTCACTCTGTCTGT R:GACCTGCCACTGTCTTCAGC	301	60	Chin et al. [36]	Chemokine attracts innate immune cells
MHC I	F:GGCTGTTTTTGCCGCTCTG R:GTGGACAGGTCTGGATAAAG	112	60	Chin et al. [36]	Molecule-presenting antigen for CD8+
β-actin (control)	F:TACCACCGGTATCGTCATGGA R:CCACGCTCTGTCAGGATCTTC	126	60	Chin et al. [36]	Reference gene

**Table 4 vaccines-09-00368-t004:** Proximate composition. Asterisks indicate a significant difference (*p* < 0.05) from the control.

Composition	Feed	
	Control	Vaccinated
Crude protein (%)	32 ± 7.2	33.1 ± 6.0
Crude lipid (%)	3 ± 1.0	4 ± 0.5
Crude fiber (%)	6.8 ± 2	6.2 ± 0.0
Ash (%)	11.6 ± 1.8	10.8 ± 2.11
Moisture (%)	12 ± 3.3	13 ± 1.4 *
Carbohydrate (%)	34.6 ± 7.65	32.9 ± 5.0

**Table 5 vaccines-09-00368-t005:** Growth performance parameters (*n* = 25 per treatment) of Asian seabass fed with vaccinated feed for 6 weeks. Asterisks indicate a significant difference (*p* < 0.05) from the control.

Parameters	Treatment Group	
	Control Group	Vaccinated Group
Initial body weight (g)	2.31 ± 0.08	2.31 ± 0.08
Final body weight (g)	9.78 ± 0.05	17.5 ± 2.23 *
Weight gain (g)	7.63 ± 1.11	14.35 ± 1.09 *
SGR (%/day)	0.36 ± 0.03	0.71 ± 0.05 *
FCR (g/g)	0.28 ± 0.06	0.16 ± 0.01 *

**Table 6 vaccines-09-00368-t006:** Details of experimental design, vaccination scheme, challenge doses, and fish survival.

Group	Number of Fish	Primary Vaccination (Day 0–5, Oral, 5% Fish Body Weight)	Booster Dose (Day 14–18, Oral, 5% Fish Body Weight	Challenge Group (10 Fish/Tank in Duplicates)	Challenge Dose/Fish (Day 28, IP, 0.1 mL/Fish)	RPS (%)
Control	80	PBS + POA	PBS + POA	Control (+PBS)	PBS	-
				Control (+Vh)	10^7^ CFU Vh	-
				Control (+Sa)	10^7^ CFU Sa	-
				Control (+Ah)	10^7^ CFU Ah	-
Vaccinated	80	10^6^ cells/kg of feed + POA	10^6^ cells/kg of feed + POA	Vaccinated (+PBS)	PBS	-
				Vaccinated (+Vh)	10^7^ CFU Vh	75 ± 7.07
				Vaccinated (+Sa)	10^7^ CFU Sa	80 ± 0
				Vaccinated (+Ah)	10^7^ CFU Ah	80 ± 0

Vh, *Vibrio harveyi*; Sa, *agalactiae*; Ah, *Aeromonas hydrophila*; POA, palm oil adjuvant; RPS, relative percent survival; -, not applicable.

## Data Availability

The data that support the findings of the study are available on request from the corresponding author.

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
