# Peer review of "Vaccine Efficacy of a Newly Developed Feed-Based Whole-Cell Polyvalent Vaccine against Vibriosis, Streptococcosis and Motile Aeromonad Septicemia in Asian Seabass, Lates calcarifer"

_vaccines, 2021, doi:10.3390/vaccines9040368_

Round 1

Reviewer 1 Report

Dear authors

The manuscript submitted to Vaccines is suitable for publication after major revision.

For me, the main problem in the manuscript is the absence of data corresponding to monovalent vaccines.  Were the formulations indicated proved as monovalent vaccines? It is possible the existence of interferences between them ?

The immunological response of the fish were different in time depending on the bacteria analyzed. My question is: how many time are the fish protected against the three pathogens?

The protection of the fish in RPS terms were not evaluated. Why?

Author Response

Cover letter

On behalf of all the authors I, Aslah Mohamad states that there is no conflict of interest about the study submitted to the journal for possible publication.

Many thanks

Sequel to your mail to us on the review of our manuscript title “Vaccine efficacy of a newly developed feed-based whole-cell polyvalent vaccine against vibriosis, streptococcosis and motile aeromonad septicemia in Asian seabass, Lates calcarifer” the following correction and rebuttals were made:

Comments and suggestions from the reviewer (Reviewer 1)

 1) For me, the main problem in the manuscript is the absence of data corresponding to monovalent vaccines. Were the formulations indicated proved as monovalent vaccines? It is possible the existence of interferences between them?

  • Thank you for your comments. For monovalent vaccine, we already developed and tested previously as being mentioned in sub-point 2.3 “In this study, we used a specific strain, harveyi strain Vh1, from our local isolates that had been shown to induce strong antigenic responses on its homologous OMPs antigen and cross-reacted against heterologous OMPs antigens of V. parahaemolyticus strain VPK1, V. alginolyticus strain VA2, and Photobacterium damselae strain PDS1 at a molecular weight of 32kDA. The strains were later prepared as an inactivated vaccine using 0.5% formalin and were tested for an in-vivo challenge using Artemia sp as a host model. The inactivated vaccine was found to give protection up to 83.3% against wild type V. harveyi [35]. In comparison, Streptococcus agalactiae strain Sa2k used in this study was previously used as an antigen in a feed-based vaccine against streptococcusis in red tilapia, Oreochromis niloticus and was found to give 100% survival against S. agalactiae [30]. Nevertheless, A. hydrophila strain Ah1sa5 used in this study was reported to be used as an antigen in a recombinant cells vaccine and was found to give 100% survival against wild type A. hydrophila in African catfish, Clarias gariepinus [33]. However, in this current manuscript, we want to explore the preliminary effects of the combinations of the different antigens in terms of enhancement or reduction in the fish immune response and others. Further study will be conducted to understand more about the mechanisms that may exist between each strain.

2) The immunological response of the fish were different in time depending on the bacteria analyzed. My question is: how many time are the fish protected against the three pathogens?

The protection of the fish in RPS terms were not evaluated.Why?

  • Yes, we agreed on experimental challenges to be included in this manuscript. The experimental challenge was done along the vaccine development process as we are aware that it is an important parameter to determine the vaccine efficacy. However, the data was kept for other publications to focus on the protective efficacy in a larger scope. However, we understand the experimental challenge to be included in this current manuscript. Thus, we decided to include the experimental challenge results in this manuscript. The experimental challenge’s materials and methodology were inserted in point 2.6.4 and the efficacy results in point 3.5.

Reviewer 2 Report

This manuscript did not conduct the most important essential test, which is a challenge test that is showing the efficacy of the vaccine. The authors have a good thinking to develop oral multiple vaccine if they are working. The authors measured several parameters that were needed to make a nice vaccine, such as IgM antibody levels that were usually not matched with protection levels, immune-related genes that had no meaning when the vaccine didn’t working, Moreover, the authors wrote the title of this paper as “ Vaccine efficacy of a newly developed feed-based whole-cell polyvalent vaccine against vibriosis, streptococcosis and motile aeromonad septicemia in Asian seabass, Lates calcarifer”, unfortunately there were no any evidences that proved the efficacy of vaccine at the essential level. This is the reason why this paper completed their vaccine study.

Author Response

Cover letter

On behalf of all the authors I, Aslah Mohamad states that there is no conflict of interest about the study submitted to the journal for possible publication.

Many thanks

Sequel to your mail to us on the review of our manuscript title “Vaccine efficacy of a newly developed feed-based whole-cell polyvalent vaccine against vibriosis, streptococcosis and motile aeromonad septicemia in Asian seabass, Lates calcarifer” the following correction and rebuttals were made:

Comments and suggestions from the reviewer (Reviewer 2)

1) This manuscript did not conduct the most important essential test, which is a challenge test that is showing the efficacy of thevaccine. The authors have a good thinking to develop oral multiple vaccine if they are working. The authors measured several parameters that were needed to make a nice vaccine, such as IgM antibody levels that were usually not matched with protection levels, immune-related genes that had no meaning when the vaccine didn’t working, Moreover, the authors wrote the title of this paper as “ Vaccine efficacy of a newly developed feed-based whole-cell polyvalent vaccine againstvibriosis, streptococcosis and motile aeromonad septicemiain Asian seabass, Lates calcarifer”, unfortunately there were no any evidences that proved the efficacy of vaccine at the essential level. This is the reason why this paper completed their vaccine study.

  • Yes, we agreed on experimental challenges to be included in this manuscript. The experimental challenge was done along the vaccine development process as we are aware that it is an important parameter to determine the vaccine efficacy. However, the data was kept for other publications to focus on the protective efficacy in a larger scope. However, we understand the experimental challenge to be included in this current manuscript. Thus, we decided to include the experimental challenge results in this manuscript. The experimental challenge’s materials and methodology were inserted in point 2.6.4 and the efficacy results in point 3.5.

Reviewer 3 Report

The Manuscript ID vaccines-1117715 entitled "Vaccine efficacy of a newly developed feed-based whole-cell polyvalent vaccine against vibriosis, streptococcosis and motile aeromonad septicemia in Asian seabass, Lates calcarifer", submitted for publication in Vaccines, deals with an interesting topic in aquaculture such as the benefits of using oral vaccination to protect fish against pathogenic bacteria as an alternative to the use of antibiotics to reduce the economic losses caused by these infections.

Although the results obtained are interesting and deserve to be published, in the reviewer opinion the work has two weak points:

1) The follow-up period should be longer, especially considering that normally the objective of vaccination is to achieve long-term protection, for which would be very interesting to see the evolution of antibody production beyond 6 weeks,

2) Although the results obtained show that the oral vaccine used induces both an innate and an adaptive immune response (production of specific antibodies), to ensure adequate protection experimental challenges should be carried out and compare the survival rate against the control group.

Other considerations and minor issues:

- Abstract: In relation to antibody levels it is stated that they showed a significant increase as early as week 2 and reached the maximum at week 6, but this is not true in the case of Ig M levels against A. hydrophila.

- Page 2, at the end of the first paragraph appropriated references should be added that support the given information.

- Page 3, in paragraph 3 a previous work is mentioned in which the authors successfully developed an oral vaccine against vibriosis, if the results are published the reference should be included.

- Page 4, Table 1: if there is any published data on the bacterial isolates used then appropriate citations should be added.

- Page 4: The reviewer assumes that all the experimental protocols (especially the infections described in section 2.2) were approved by a Bioethics Committee and should therefore be cited in the manuscript.

- Page 6, section 2.4.4: why the growth performance test was carried out with a batch of fish other than the one used in vaccination trial? It should be clearly stated that the growth performance and immune response assays were performed on different fish groups.

- Page 6, section 2.5.2: Why did you decide to give the vaccine for 5 consecutive days? It is important to justify this regimen because it has been reported that the efficacy of oral vaccination in fish depends largely on the dose and the administration regimen (Mutoloki et al. 2015, Front Immunol, doi: 10.3389/fimmu.2015.00519).

- Page 7, section 2.6.1: review the ELISA method described because it is stated that conjugated anti-goat were used but the primary antibody was from mouse (mouse anti-Asian seabass).

- Page 9, the section 3.1 title should be include growth performance as well “3.1. Feed quality analysis and growth performance”.

- Page 9, section 3.1: the text “The figures demonstrate that the group from commercial formulated vaccinated feed shows higher physical stability in the water as compared to the control group, commercial feed” should be deleted because the difference observed were not significant.

- Page 9, section 3.1: The growth performance results were presented in Figure 4 and Table 5, not in table 4 as is stated in the text.

- In all Figures (except 1 and 2) should be indicated that the results are expressed as mean values, standard deviation and the number of determinations (n) in the same way that it is in Figure 8.

- In Table 5 figure that the number of fish used were 30 but in M&M section it was mentioned that 50 fish were used, so it should be 25 fish per group or treatment.

- In discussion section the results obtained about antibody response and the relative expression of the different genes related with immune response should be interpreted and compare with those obtained by others authors.

- The conclusion should be reformulated because with the results presented in the manuscript it cannot be concluded that it reduces the infection rate and protects the vaccinated fish (this could only be demonstrated if experimental infections were carried out).

- References: check references 23 and 24, and 27 and 28 because they seem to be the same.

- Supplementary information: the given information is exactly the same as the five tables included in the manuscript.

Author Response

Cover letter

On behalf of all the authors I, Aslah Mohamad states that there is no conflict of interest about the study submitted to the journal for possible publication.

Many thanks

Sequel to your mail to us on the review of our manuscript title “Vaccine efficacy of a newly developed feed-based whole-cell polyvalent vaccine against vibriosis, streptococcosis and motile aeromonad septicemia in Asian seabass, Lates calcarifer” the following correction and rebuttals were made:

Comments and suggestions from the reviewer (Reviewer 3)

General comments

1) The follow-up period should be longer, especially considering that normally the objective of vaccination is to achieve long-term protection, for which would be very interesting to see the

evolution of antibody production beyond 6 weeks.

  • We agreed that the follow-up period should be longer to see long-term protection. However, in this paper, we want to see the preliminary effects of the vaccination using the newly developed feed-based polyvalent vaccine in terms of the development process, feed quality after vaccine incorporated with the powdered feed and short terms innate and adaptive immune response post vaccinations. Yet, a shorter preliminary vaccine period was observed in Mao et al. [1] with just 28 days vaccination period and Cao et al. [2] with 30 days vaccination period. A similar time frame (6 weeks) was observed in Hoare et al. [3] and Laith et al. [4]. Nevertheless, future research will be conducted in a longer vaccination period.

2) Although the results obtained show that the oral vaccine used induces both an innate and an adaptive immune response (production of specific antibodies), to ensure adequate protection

experimental challenges should be carried out and compare the survival rate against the control group.

  • Yes, we also agreed on experimental challenges to be included in this manuscript. The experimental challenge was done along the vaccine development process as we are aware that it is an important parameter to determine the vaccine efficacy. However, the data was kept for other publications to focus on the protective efficacy in a larger scope. However, we understand the experimental challenge to be included in this current manuscript. Thus, we decided to include the experimental challenge results in this manuscript. The experimental challenge’s materials and methodology were inserted in point 2.6.4 and the efficacy results in point 3.5.

Other considerations and minor issues:

3) Abstract: In relation to antibody levels it is stated that they showed a significant increase as early as week 2 and reached the maximum at week 6, but this is not true in the case of Ig M

levels against A. hydrophila.

  • Thank you for your comments. We apologies for the conflict and we corrected inside the abstract in line 18-21 where we mentioned, “Following vaccination by the polyvalent feed-based vaccine, IgM antibody levels showed a significant (P < 0.05) increase in serum against Vibrio harveyi, Aeromonas hydrophila and Streptococcus agalactiae and reached the peaked at week 3, 5 and 6, respectively. The high stimulated specific antibody in the serum remained significantly higher than control (P < 0.05) at the end of the six-week vaccination trial”.

4) Page 2, at the end of the first paragraph appropriated references should be added that support the given information.

  • References [5, 6] were added to support the given information.

5) Page 3, in paragraph 3 a previous work is mentioned in which the authors successfully developed an oral vaccine against vibriosis, if the results are published the reference should be

included.

  • The results are yet still unpublished. Hence, citation as unpublished data is included in Line 128.

6) Page 4, Table 1: if there is any published data on the bacterial isolates used then appropriate citations should be added.

  • Appropriate citations in Table 1 were included.

7) Page 4: The reviewer assumes that all the experimental protocols (especially the infections described in section 2.2) were approved by a Bioethics Committee and should therefore

be cited in the manuscript.

  • The experimental protocols were approved by the bioethics committee and is cited in Line 173-175.

8) Page 6, section 2.4.4: why the growth performance test was carried out with a batch of fish other than the one used in vaccination trial? It should be clearly stated that the growth performance and immune response assays were performed on different fish groups.

  • Similar fish was used for the growth performance parameters. Section 2.4.4 was re-write and was put under a new sub-topic in section 2.5.4.

9) Page 6, section 2.5.2: Why did you decide to give the vaccine for 5 consecutive days? It is important to justify this regimen because it has been reported that the efficacy of oral vaccination

in fish depends largely on the dose and the administration regimen (Mutoloki et al. 2015, Front Immunol, doi:10.3389/fimmu.2015.00519).

  • We decided to give the vaccine for five consecutive days based on Mao et al. [1], Samuelsen, [7], Monir et al. [8], Sun et al. [9], Sun et al. [10], Tu et al. [11] and, Wang et al. [12]. The researchers studied oral vaccine in fish and gave the oral vaccines to the fish for five consecutive days. Five days was thought to be suitable for the oral vaccination considering the antigen structure that may be destroyed by the fish stomach’s gastric conditions at the early days of vaccinations. However, as this is a preliminary study on the effect of the polyvalent feed-based vaccine, further study will be conducted with shorter or longer days of vaccination to find the most optimal duration for oral vaccinations in terms of efficacy and cost.
  •  

10) Page 7, section 2.6.1: review the ELISA method described because it is stated that conjugated anti-goat were used but the primary antibody was from mouse (mouse anti-Asian seabass).

  • Thank you for your comments regarding this matter. The ELISA method was corrected and further explained in Line 303-306.

11) Page 9, the section 3.1 title should be include growth performance as well “3.1. Feed quality analysis and growth performance”.

  • Yes, we agreed that the section 3.1 title should be include growth performance as well. The changes had been made in the tittle section 3.1.

12) Page 9, section 3.1: the text “The figures demonstrate that the group from commercial formulated vaccinated feed shows higher physical stability in the water as compared to the control group, commercial feed” should be deleted because the difference observed were not significant.

  • Yes, we agreed that the text “The figures demonstrate that the group from commercial formulated vaccinated feed shows higher physical stability in the water than the control group, commercial feed” should be deleted because the difference observed was not significant and should be deleted.

13) Page 9, section 3.1: The growth performance results were presented in Figure 4 and Table 5, not in table 4 as is stated in the text.

  • Yes, we agreed that the growth performance results were presented in Figure 4 and Table 5, not in table 4, as is stated in the text. Thank you for your comments and apologies for the conflict.

14) In all Figures (except 1 and 2) should be indicated that the results are expressed as mean values, standard deviation and the number of determinations (n) in the same way that it is in

Figure 8.

  • Yes, we agreed that all figures (except 1 and 2) should be indicated that the results are expressed as mean values, standard deviation and the number of determinations (n) in the same way that it is in Figure 8. The indications were inserted in Figure 3-7.

15) In Table 5 figure that the number of fish used were 30 but in M&M section it was mentioned that 50 fish were used, so it should be 25 fish per group or treatment.

  • Yes indeed. We appreciate your comment on improving this manuscript. The number of fish used in Table 5 figure should be 25 fish per group.

16) In discussion section the results obtained about antibody response and the relative expression of the different genes related with immune response should be interpreted and compare with those obtained by other authors.

  • Thank you for your comments. The different genes related to immune response was interpreted and compared with those obtained by other authors in the discussion section in Line 550 - 560.

17) The conclusion should be reformulated because with the results presented in the manuscript it cannot be concluded that it reduces the infection rate and protects the vaccinated fish (this

could only be demonstrated if experimental infections were carried out).

  • As the experimental challenge is already included in the manuscript, the conclusion shall be retained.

18) References: check references 23 and 24, and 27 and 28 because they seem to be the same.

  • Thank you for your comments on improving this manuscript. One of each duplicate was removed from the reference list.

19) Supplementary information: the given information is exactly the same as the five tables included in the manuscript.

  • Yes indeed. We are aware of the similarities; thus, supplementary information such as raw data and others will be included upon request.

Round 2

Reviewer 1 Report

Dear authors

The manuscript was improved following the recomendations of the reviewer.

The authors included in the manuscript experimental in vivo assays. However I consider that the dose inoculated are not the correct. The mortalities in the control group are very high and this may be due to the high dose inoculated: 107. The legislation applicable to vaccination assays in fish ((European Pharmacopeia. Pharmeuropa 23.1. 2011 pp122-123,  Guideline on the design of studies to evaluate the safety and efficacy of fish vaccines EMA/CVMP/IWP/314550/2010) recommend that the dose inoculated should be approximately the LD60 of the pathogen. In base to this the RPS value are overrated.

Moreover in Table 6. the data are not correct. The survival in the control group challenged is =0? this data should be 100%.

All these data and experiments should be corrected.

Author Response

Dear reviewer, 

Thank you. 

Reviewer 2 Report

This manuscript makes me be confused about the challenge test results that showed survival %. Since it is usual to present as a RPS to uderstand the efficacy of a vaccine. Another thing is that I don't have much idea of Asian seabass, Lates calcarifer, especially immune resposne that exhibited very higher immune response in terms of survival percentage.  As far as I know it is not easy to reach such a high survival percentage, if the isolates challgened was weak virulance. Can you describe the two above questions? 

Author Response

Dear reviewer, 

Thank you. 

Reviewer 3 Report

The reviewer considers that the quality of the manuscript has been increased in this revised version especially by including the survival results after the experimental challenge.

However, he would like to make the following comments to the authors:

In the first place there is great confusion concern to the references that are cited in the text sometimes with the name of the first author and other times with a number. Furthermore, the citations of the text do not coincide with those indicated in the reference list, so it is impossible to find the appropriate corresponding one. For example, in Table 1 a reference with the number 33 is cited, which should correspond to the published data of the A. hydrophila strain used in the present work and however in the reference list this number is a paper on vaccination strategies against to bluetongue? The same for the reference given in lines 268-269 and many others throughout the manuscript.

Authors said that similar fish was used for the growth performance test and vaccination trial, I suppose they refer to the same species of fish because it is clear that different batches of fish were used because those used in the vaccination had an average of 15 g (see section 2.5.1) and those used in the growth test 2 gr as you can check clearly from the data represented in figure 4.

In section 2.6.4 it is stated that the experimental infection was tested 28 days post-immunization; But after the first vaccination in week 0 or after the booster in week 2? I also consider that this information (time of the challenge) could be included in figure 1, which would improve the understanding of the experimental design.

In Figures 5-9 it is shown that the number of fish sampled was 6 but in section 2.5.2 it is said that the samples were taken from 5 fish each week.

The data presented in Table 6 should be reviewed since it appears that the survival of the control group (unvaccinated and uninfected) was 0%.

Author Response

Dear reviewer, 

Thank you. 

Round 3

Reviewer 2 Report

Oral vaccination is definitely important in terms of fish vaccination. Some of the data referred were not published that would be needed to prove the confidence. If the authors had more previous data, it should be better

Author Response

Dear reviewer, 

Please kindly see the attachment
